# Photothermal Effect in Plasmonic Nanotip for LSPR Sensing

**DOI:** 10.3390/s20030671

**Published:** 2020-01-25

**Authors:** Muhammad Shemyal Nisar, Siyu Kang, Xiangwei Zhao

**Affiliations:** 1State Key Laboratory of Bioelectronics, School of Biological Science and Medical Engineering, Southeast University, Nanjing 210096, China; mshemyalnisar@hust.edu.cn (M.S.N.); 220191928@seu.edu.cn (S.K.); 2Southeast University-Shenzhen Research Institute, Shenzhen 518000, China

**Keywords:** optical fiber sensors, localized surface plasmon resonance, nanotip, electromagnetic heating, refractive index

## Abstract

The influence of heat generation on the conventional process of LSPR based sensing has not been explored thus far. Therefore, a need exists to draw attention toward the heat generation issue during LSPR sensing as it may affect the refractive index of the analyte, leading to incorrect sensory conclusions. This manuscript addresses the connection between the photo-thermal effect and LSPR. We numerically analyzed the heat performance of a gold cladded nanotip. The numerical results predict a change in the micro-scale temperature in the microenvironment near the nanotip. These numerical results predict a temperature increase of more than 20 K near the apex of the nanotip, which depends on numerous factors including the input optical power and the diameter of the fiber. We analytically show that this change in the temperature influences a change in the refractive index of the microenvironment in the vicinity of the nanotip. In accordance with our numerical and analytical findings, we experimentally show an LSPR shift induced by a change in the input power of the source. We believe that our work will bring the importance of temperature dependence in nanotip based LSPR sensing to the fore.

## 1. Introduction

When light interacts with the conduction electrons of a metal at the metal–dielectric interface, it brings about coherent oscillations of these electrons. These excited oscillating free electrons are called surface plasmons. The field of research concerned with the study of this light–metal interaction is called ‘plasmonics’, which is part of a larger field of nanophotonics [1]. Plasmonics has found numerous applications in the fields such as Surface Enhanced Raman Scattering (SERS) [2,3,4,5], Tip Enhanced Raman Scattering (TERS) [6,7,8], Negative Index of Refraction (NIR) [9,10], solar cells [3,11], Localized Surface Plasmon Resonance (LSPR), etc. Techniques such as LSPR, TERS, and SERS have been extensively used for label-free sensing. LSPR, the exponentially decaying plasmon produced by electric field traveling along the metal interface, is an important technique for label-free sensing of chemical and material information at the nanoscale [12]. The spectrum of LSPR at the metal–medium interface depends on various parameters such as the morphology of the metallic structure, irradiating electromagnetic field, and the refractive index of the environment [13]. This opens up the possibility of tuning the response of these structures or their use as sensors through their ability to discriminate between refractive indices. Better ability to discriminate between consecutive refractive indices is reflected in the sensitivity of the sensor, which, among other things, also depends on the morphology of the metallic structure [13].

In recent times, LSPR has been instrumental for numerous sensing applications relating to photochemistry [14], electrochemistry [15,16], material discernment and identification [2,17,18], concentration sensing [19,20], acidity sensing [21], trace element detection [22], and intercellular processes [23,24,25]. Broadly, these applications use two methods to obtain the LSPR. A vast amount of available literature revolves around using nanoparticles in various morphologies to obtain LSPR [20,26,27]; on the other hand, there is some research on the use of planner structures of variously patterned metal films to obtain LSPR from these metal films [15,28,29]. However, these methods cannot produce surface plasmons at a localized, specific location.

In an attempt to attain a small localized resolution, attempts have been made to equip optical fibers with nano-antennas. These studies can be categorized into three groups: (i) Those that use a geometry modified optical fiber [30,31,32], (ii) those that equip facets of a fiber with nano antenna [32,33,34,35,36], and most recently (iii) those that use a tapered fiber [37,38]. These studies [37,38] use tapered fiber tips as TERS tips. A recent study [39], acknowledging this gap in the research, presented a cholesterol sensor that used nanoparticles attached to a tapered optical fiber tip. These nanoparticles produced LSPR, which was then analyzed through standard spectrometry by using the spectral shift in it as their unit of analysis.

As the LSPR spectrum depends on the refractive index of the environment [1], all of these methods operate by being able to distinguish subtle changes in the refractive index of the environment through the associated changes in the LSPR. These sensors are reportedly to be so sensitive that they can even distinguish a refractive index change of about 0.001 RIU [31]. The dependence of the refractive index on temperature and the input wavelength of the incoming electromagnetic wave [40,41] is well understood. The phenomenon holds true for organic as well as inorganic materials such as silicon and germanium [42]. As a result, it is can be a priori inferred that changes in the temperature of the environment of the medium would change the refractive index of the medium. This change in the refractive index should be observable in the LSPR spectrum obtained from our instruments.

Photo-thermal heating is a well-established physical phenomenon that manifests as an increase in temperature in the micrometer region around the metal–medium interface [43,44,45,46]. Metallic nanostructures are known to generate heat when they are illuminated at the wavelength of their plasmonic resonance. The process of heat generation depends on the morphology of the metallic structure and the incident wavelength [47]. The heat generation is due to the losses that these surface plasmons travelling along the metal–medium interface encounter in their propagation [48,49]. These losses are known to limit the propagation length of the surface plasmons, thus making them localized, and heating the surrounding medium. The heating of the plasmonic structures has opened up various rapidly expanding research areas collectively known as thermoplasmonics [50]. Recently, numerous studies have sought to use thermoplasmonics for a wide variety of applications. These applications include photovoltaics [51,52], liquid heating [53,54], gene therapy [55], thermal biology [56], photothermal cancer therapy [57,58,59,60], imaging and spectroscopy [61,62], and plasmofluidics [63,64]. Some of the thermoplasmonic studies have used the thermally caused shift in the LSPR as a mechanism of temperature sensing [65], whereas Jackman et al. [66] devised a highly sensitive method using LSPR that analyzes the deformation in the shape of the protein molecule as it is adsorbs on a metal surface at different temperatures. However, most of the works that connect LSPR and the temperature of the analyte, do it for bulk temperature and do not consider micro/nano level temperature in the vicinity of plasmonic structure.

Given the inherent conflict among the results from these two different kinds of studies, it can be postulated that the use of LSPR as a sensing mechanism requires extreme care. This is because it is prone to error due to the change of the refractive index of the analyte through heat generation from surface plasmons. Thus, a need exists to draw attention toward the heat generation issue during the LSPR sensing process.

In this manuscript, we analyzed the heat performance and LSPR from a gold cladded tapered optical fiber tip used as a nanotip. We numerically simulated the fiber tip to show that the transmission of light in the tapered fiber leads to an increase in the temperature of the micro region near the tip. We subsequently analytically calculated the resulting change in the refractive index of the surrounding nano/micro medium. This paper then presents an experimental demonstration of the shift in LSPR with a change in the intensity of the source when used with transmission spectroscopy (intensity power relation). This paper then attempts to link the shift in LSPR with respect to the intensity of the source with the increase in temperature due to the photo-thermal effect. Finally, we comment on the reliability of LSPR from a tapered nanotip as a sensing method under transmission spectroscopy.

## 2. Materials and Methods

### 2.1. Simulation Design

To simulate the nanotip, a Finite Element Method (FEM) based multi-physics simulation was setup on a commercially available software (the details of the simulation setup are provided in the Appendix A, along with Appendix A detailing important parameters used in the simulation setup). As a nanotip is axially symmetric, a 2D-axisymmetric simulation design was constructed in the interest of computational resources and the time requirement for such a simulation. The simulation setup aims to calculate the electromagnetic propagation in a gold cladded nanotip and then link the propagating and plasmonic modes in the tip to heat the transfer module. The complete schematic of the simulation setup along with the details with regards to mesh are available in Appendix A, given in the Appendix A.

### 2.2. Experimental Design

In order to experimentally verify this hypothesis, the experiment was set up as shown in Figure 1. A light source from Ocean Optics (DH-2000) with an output power of 950 µW was set up. The light source used by us was the same as that used by [31,36]. The output from the light source was fed into an optical fiber that terminated at a Neutral Density Filter (ND-filter). For the purpose of the experiment, ND filters with values of 80, 50, 25, and 10 were used, whereas the 100 percent power could be obtained by the removal of the filter altogether. This ND-filter was then connected to another multimode fiber that terminated at a gold cladded nanotip procured from Rayme Biotechnology. The schematic and Scanning Electron Microscope (SEM) picture of the nanotip are given in the insets in Figure 1.

The nanotip was in-turn immersed in the analytes such as methanol, ethanol, IPA (iso-propyl alcohol/propanol) and deionized water. While deionized water was laboratory produced, the other chemicals were procured from the Sinopharm Chemical Reagent Company. The light in the analyte was collected using an objective lens of an Olympus IX73 microscope and analyzed using SpectraPro HRS-500 from Princeton Instruments with a spectral resolution of 0.1 nm (i1-120-500-P Ruled grating, 68 x 68 mm, 1200 G/mm with 500 nm blaze wavelength). The working distance between the objective and the tip was maintained at around 2 cm.

## 3. Results and Discussion

The electromagnetic output from the nanotip is given in Figure 2a. The electric field intensity should exponentially decrease as a function of distance from the apex, as shown in Figure 2c. Given that our tip has a diameter of 300 nm at the apex, it was sufficient to support some of the modes of the visible spectrum, therefore, a relatively high electric field intensity just below the silica–air interface can be observed from the inset given in Figure 2a. To further confirm this, modal analysis was done on the tip and it was observed that a number of modes of the visible regime could be accommodated in the apex. The profile of one such mode is given in the inset of Figure 2b, while the real and imaginary parts of the effective index of this particular mode are given in Figure 2b. The effective index plot shows a significant difference in the real effective index of the mode at a lower frequency range compared to that at the high frequency range.

With the confirmation of the optical response from our gold cladded nanotip, we proceeded to check its heat response. Our FEM based computation package computes the heat response in the time domain. Heat is generated at the metal interfaces as this is the location where the surface plasmons are generated.

Figure 3c shows that the system reached a steady state temperature (which is in equilibrium with the environment) in less than 100 milliseconds. The temperature profile of the tip is shown in Figure 3a,b at t = 0 s and at t = 0.1 s, respectively, when the system is in a steady state. As common sense dictates and can be observed in Figure 3c,d, the temperature decreases exponentially as we move away from the apex in the medium. The profile of the steady state temperature is similar to that reported in the literature [54,67]. The difference between the steady state temperature and the initial temperature at the apex, ΔT, is approximately 14 K. This difference reduces as we move away from the apex and at the distance of 34 μm, the ΔT is merely 2 K, which would reduce to zero as we move even further away.

Dependence of the refractive index on the temperature of the medium is a well understood phenomenon [40,41]. It is also evident from the corpus of literature available on the subject, that the LSPR depends on the refractive index of the medium [31,68]. Therefore, we used already published data to analytically calculate the changes in the refractive indices of some small organic molecules that are liquid (at room temperature) and water around the temperature range that our simulation results predicted. These chemicals were methanol, water, acetone, ethanol, and propanol with the refractive indices of 1.3395 [69], 1.3337 [70], 1.3614 [71], 1.3637 [71], and 1.3794 [72], respectively, for the incident wavelength of 532 nm (the relationship of the wavelength and refractive index of these chemicals is provided in the Appendix A) at room temperature. The rate of change of the refractive index of methanol, ethanol, acetone, and propanol is −4 × 10^−4^ RIU (Refractive Index Unit) per Kelvin [40], whereas the rate of change of the refractive index of water with respect to temperature is approximately −5.2 × 10^−5^ RIU per Kelvin [41]. With these values, the refractive index of these chemicals at an increasing distance from the apex is shown in Figure 4a. As the rate of change of the refractive index of water per unit change in temperature is an order of magnitude smaller than the organic chemicals, the refractive index of methanol and water crossed-over at a certain distance from the apex and then diverged beyond this point with respect to distance. As the LSPR depends on the refractive index, and the refractive index depends on the temperature; it can, therefore, be deduced that the LSPR also depends on the micro-scale temperature around the tip, making temperature an important variable.

The available literature tells us that the change in temperature is caused by surface plasmons and the intensity of surface plasmons depends on the intensity of the incident wave [73,74,75]. This means that the change in temperature should also depend on the change in the intensity of the incident wave. The numerical simulations show (Figure 4b) that the micro temperature of the medium also changes due to changes in the input power. The linear relationship between the input power and the temperature at the apex can be seen in Figure 4c. This, therefore, implies that the change in our input power should provide us with a shift in the LSPR, as the change in input power would bring about a change in the temperature of the micro-environment, and this change in temperature influences the LSPR.

After verification through numerical and analytical methods, we finally sought to experimentally verify the theoretical claims. With the experimental setup arranged as given in Figure 1, the experiment was carried out with methanol, ethanol, IPA (iso-propyl alcohol/propanol), and deionized water and the intensity of the incident light was controlled by the introduction of ND-filters in the optical pathway. The results of these experiments are given in Figure 5a. Due to the limitations of the experimental design of the transmission spectroscopy, the reduction in the power led to an increased contribution of noise in the received spectrum. When using the filter of 10 percent, the contribution of the noise increased to such an extent that it became difficult to separate the noise from the received spectrum. Due to this reason, the values were removed from the results shown in Figure 5a. These results are nevertheless provided in the Appendix A.

In order to further validate that the origin of the effect was due to LSPR, a control experiment was carried out using a tapered uncladded fiber immersed in IPA. The results of the control experiment are provided in Figure 5b. As Figure 5 shows, there was no shift in the LSPR peak when a fiber tip without gold cladding was used. This confirms that the effect given in Figure 5a is due to the presence of LSPR.

After experimental verification of the shift in LSPR with respect to change in temperature mediated through change in the intensity of the incoming light, we wanted to theoretically analyze the effect on temperature in the micrometer region near the nanotip of the change in the diameter of the tip and the change in diameter of the fiber. The general trend was that the temperature exponentially decreased with increasing fiber diameter. The decrease in the temperature was linked with decreasing attenuation that is encountered by incident light in an expanding cavity as a narrower cavity is a filter for larger wavelengths. Moreover, a narrower cavity with the same amount of incident light would have more energy concentrated in a smaller region, and can also potentially lead to increased temperatures in narrower cavities. The relationship is shown in Appendix A.

Figure 5a clearly shows that there was little shift in the wavelength of the LSPR peak of deionized water. This corroborates well with the change in the refractive index of deionized water with the change in temperature, as given in Figure 4a. The insignificant change in refractive index led to an insignificant shift in the LSPR. While the rate of change of the wavelength of the LSPR peak for ethanol and IPA was very similar to each other, these results were supported by the results for the rate of change of the refractive index of both these analytes with the change in temperature (given in Figure 4a). On the other hand, the rate of change of wavelength of the LSPR peak of methanol is an order of magnitude less than that of ethanol and IPA, although the rate of change of refractive index of methanol with respect to temperature is similar to other alcohols. This difference can be attributed to the difference in the change in refractive index of these materials with respect to wavelength, as shown in Appendix A provided in the Appendix A. There was a considerable difference in the rate of change of the refractive index with respect to the wavelength between methanol and other analytes. While the rate of change of the refractive index with respect to the input wavelength was almost the same for ethanol and IPA, methanol’s curve also intersected with water, ethanol, and finally IPA in the ultra violet region of the spectrum.

It is evident that the importance of the intensity of incident light in the LSPR based sensing applications cannot be overstated. The instruments of excitation used in some important works are tabulated in Appendix A. Here, is it is pertinent to mention that, unlike the literature on thermoplasmonics, there is a dearth of information regarding the incident power used for experimentation in the literature on LSPR based sensing. The information cited in Appendix A was therefore acquired by checking the instrument used. In some cases, even the instrument itself is not mentioned and such cases were ignored in Appendix A. Nevertheless, the power used by LSPR studies is at least an order of magnitude less than the power of incident light in the thermoplasmonic studies tabulated in Appendix A. Fang et al. [54] reported that using gold nanoparticles with a diameter of 100 nm, at the incident power level of less than 25 mW, the temperature of the surrounding solution increases. When the incident power was above 25 mW, it resulted in the formation of a nanobubble surrounding the nanoparticle due to vaporization of the solution in its vicinity. Moreover, the reduction of power to 1 mW did not lead to an increase in temperature. However, Fang et. al., like most of the other studies on thermoplasmonics, used gold nanoparticles as a source of plasmonic heating. The nanoparticles of various morphologies have a smaller surface area. Due to the smaller surface area, they interact with a smaller percentage of incoming photons and smaller regions of the surrounding medium [67]. Therefore, they require higher incident power to increase the temperature. Conversely, our nanotip was orders of magnitude larger in dimension when compared with the reported nanoparticles. This leads to increased temperatures at incident power levels for which nanoparticles are thermally non-responsive. This implies that the surface area of a plasmonic device is also an important parameter to ascertain the accurate amount of heat generated by the device. This leads us to the understanding that the surface area of the plasmonic device immersed in the analyte should also be considered for an accurate calculation of the increase in temperature.

As the increase in temperature is caused by the interaction of surface plasmons with the metal [48,49,50]. This means that any situation that entails a sufficient amount of such interactions should, in principle, lead to an increase in temperature, causing a temperature based change in refractive index leading to inaccurate measurements of LSPR. For a given system, the increase in temperature (as clearly seen Figure 4b,c given above) linearly depends on the input power of the laser source. This implies that in order to obtain accurate results from LSPR based instruments, the input laser power should be low enough to not to cause a “significant” increase in the temperature. Given that our experimental design employs transmission spectroscopy, Appendix A shows that the reduction of power below a certain level makes the spectrum very noisy. In an experimental design that uses reflection spectroscopy, as in the case of [39], the resulting spectrum would be less noisy. This should easily enable them to further reduce the power. However, this privilege is not available for an experimental design that uses transmission spectroscopy, as used in this study.

In order to ascertain what can be considered as a “significant” increase in temperature, the rate of change of the refractive index with respect to temperature for the analyte also needs to be taken into account. This is because of the fact that the rate of change of the refractive index with respect to temperature is not the same for all analytes, as given in Figure 4a. This means that what is considered as a “significant” increase in temperature for a particular analyte may or may not be “significant” for other analytes. Therefore, it is also an important parameter to consider when designing a LSPR based sensing system.

## 4. Conclusions

We looked at the presence of the photothermal effect at the apex of a plasmonic tip. We explored its link with the change in the microscale refractive index in the vicinity of the nanotip. The temperature based change in the refractive index is reflected by the shift of LSPR. The experimental evidence shows that the shift in the LSPR is not constant for all analytes and is dependent on the rate of change of the refractive index of each particular analyte with respect to the change in temperature. This becomes more complicated due to the fact that the refractive index of the material is not constant for all frequencies of the electromagnetic spectrum. As a result, the temperature at the apex of the nanotip changes, and is subject to, among other things, the profile of the nanotip and the input power used. 

All of the above discussion implies that using a metal-cladded nanotip as an LSPR sensor requires extreme care as a mismatch of the physical dimensions of the nanotip, incident power, and the length of the nanotip immersed in the solution can easily lead to increase in surrounding temperature causing inaccurate LSPR based sensing. On the hand, it also opens up the possibility of using a metal cladded nanotip as a low powered nanoheater. This would provide localized heating to only the desired location and the ability to control the location that is heated, something that is not possible with the distributed nanoparticles used in most thermoplasmonic studies.

## Figures and Tables

**Figure 1 sensors-20-00671-f001:**
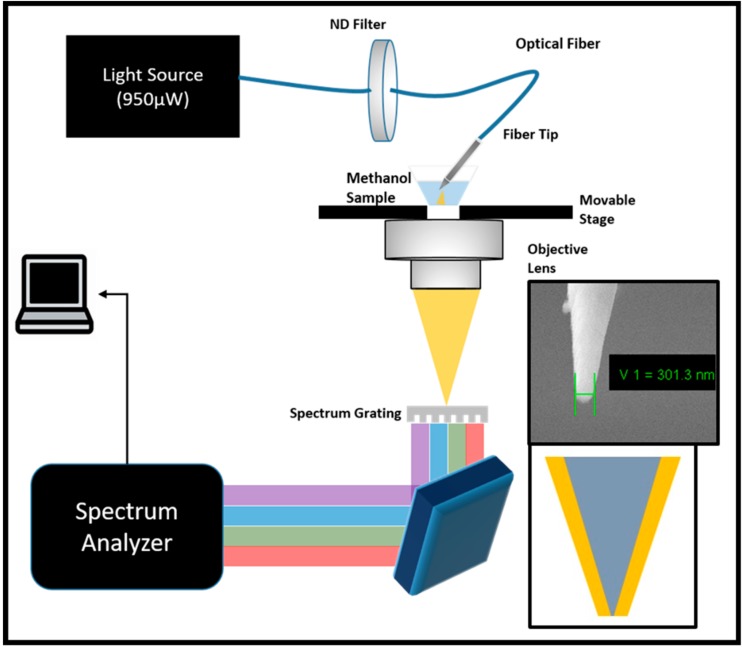
Experimental setup used in order to test the LSPR shift with change of intensity of the source. ND-filters of various gradations were used to change the intensity of light while methanol, ethanol, propanol, and water were used as samples. The light was collected from the objective and focused on the grating of the spectrometer and the spectrum was observed on the computer. The inset shows the Scanning Electron Microscope (SEM) image of the tip used for the experiments.

**Figure 2 sensors-20-00671-f002:**
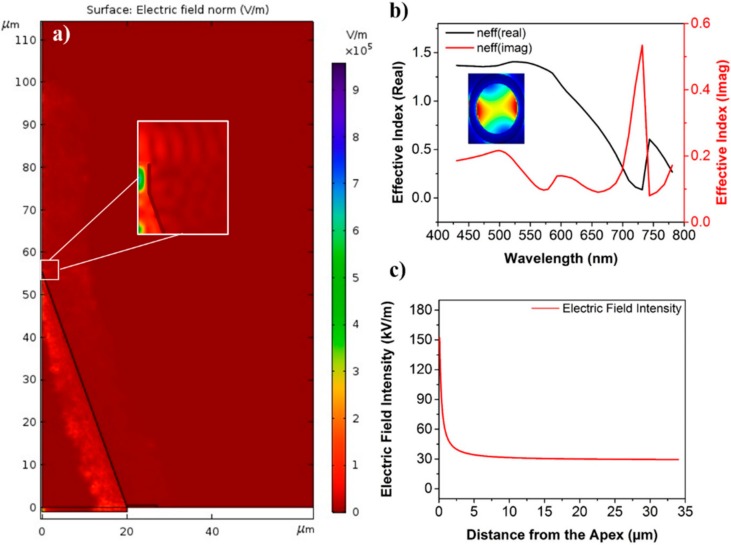
Electro-magnetic simulation of the nanotip/ (**a**) Normal electric field intensity profile inside the gold cladded nanotip; the inset shows the close-up at the apex. (**b**) The real and imaginary effective index of one of the modes in the apex of the tip is shown for the visible frequency spectrum, the particular mode profile is given in the inset. (**c**) The electric field intensity as a function of distance from the apex of the nanotip.

**Figure 3 sensors-20-00671-f003:**
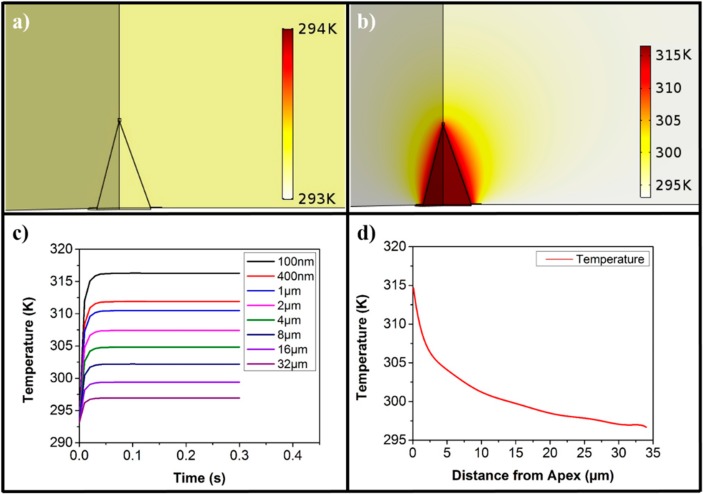
Heat response of the nanotip from the FEM based simulation. (**a**) Temperature profile of the nanotip at time t = 0 s. (**b**) Temperature profile of the nanotip at a steady state temperature using an input wavelength of 532 nm. (**c**) Temperature (in Kelvins) as a function of time at various distances from the apex of the nanotip for the input wavelength of 532 nm. (**d**) A curve fit of steady state temperature in Kelvins as a function of distance from the apex for the input wavelength of 532 nm.

**Figure 4 sensors-20-00671-f004:**
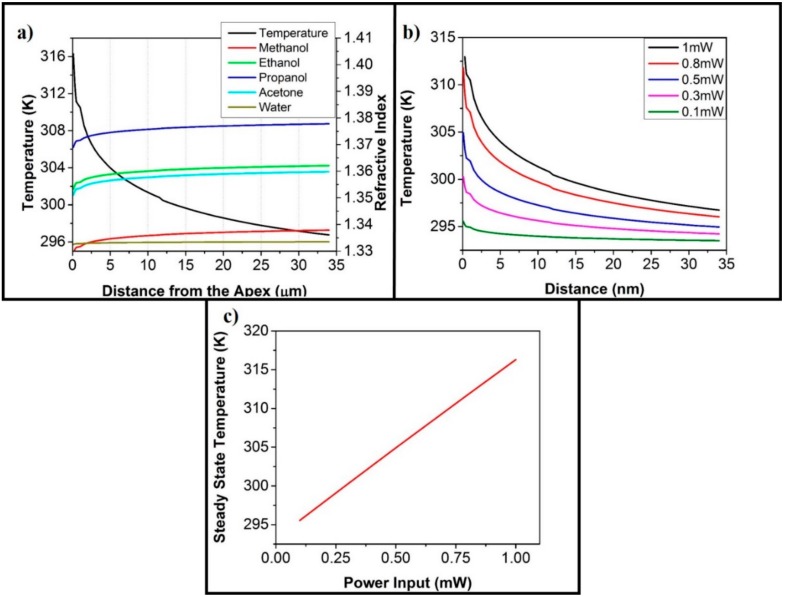
Dependence of temperature with the other parameters. (**a**) The graph shows the relationship of refractive index and the distance from the apex when using the input source of 532 nm. This change in refractive index is brought about by changing temperature with changing distance. The temperature profile along the same distance is also shown on the left y-axis to bring forth the relationship between the refractive index and the ambient temperature of the micrometer region. (**b**) The change in temperature profile with a change in the input power of the source with an input wavelength of 532 nm. (**c**) The change in temperature at the apex with the change in input power shows a linear relationship.

**Figure 5 sensors-20-00671-f005:**
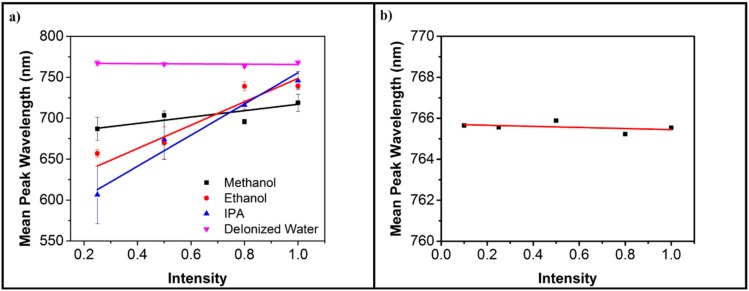
Spectrum of the experimental output. (**a**) The scatter plots of the wavelength of the LSPR peak with respect to normalized intensity for each of the analyte chemicals. The best-fit line was later added to the scatter plot to make the relationship clear. The vertical lines represent variance in the data from multiple experimental readings while the spot itself in the average value of those multiple experiments for each of the normalized intensity. (**b**) The control experiment with a tapered fiber-tip without metal cladding. The scatter plot’s linear fit is also provided.

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
