# Peer review of "Photothermal Effect in Plasmonic Nanotip for LSPR Sensing"

_sensors, 2020, doi:10.3390/s20030671_

Round 1

Reviewer 1 Report

Nisar et al. investigate photothermal effects on localized surface plasmon resonance (LSPR) sensors through a combination of analytical calculations and experimental verification. The question is interesting, however, the conclusions extend beyond what the data supports, i.e., "All of the above discussion implies that the very act of measuring the refractive index by use of LSPR actually changes the refractive index of the environment it is supposed to measure in the first place. This puts into question the whole enterprise of LSPR based instruments that are getting ubiquitous by the day for all sorts of measurements ranging from material identification to concentration measurement." These authors lump together the entire field of temperature-related LSPR studies with such general statements, however, the effects of photothermal generation are more far more nuanced depending on the light intensity and application needs. The authors need need to significantly revise the manuscript to provide a clearer discussion of these points and address the following points. With appropriate revision, I think the manuscript could be potentially publishable. 1. The simulations are presented for a very specific nanostructure (nanotip). How generalizable are these findings to other nanostructures? Please comment. 2. Please add more discussion about how the light intensity relates to the values typically used in photothermal generation vs. sensing applications. The intensity and light source is often quite different between these two application cases and please provide more detailed information, preferably a table comparing key parameters so that it is clear to the readership. 3. In addition to photothermal generation, temperature-dependent LSPR sensing experiments have been conducted and should be discussed within this article, including aspects such as temperature on signal-to-noise ratio etc. Relevant references to discuss include: DOI: 10.1021/jp409264w and DOI: 10.1021/acs.analchem.7b03921. 4. It is mentioned that "We have verified the presence of photothermal effect at the apex of a plasmonic tip. We explored its link with the change in the refractive index which is practically reflected in the shift of LSPR". This is too simplistic and bulk refractive index and surface sensitivities should be further analyzed in order to provide deeper and stronger quantitative insight.

Author Response

Prof. Dr. Xiangwei Zhao

State Key Laboratory of Bioelectronics

School of Biological Science and Medical Engineering, Southeast University

2 Si Pai Lou, Nanjing 210096, China

Tel/Fax: +86-25-83795632

15-Jan-2020

RE: Photothermal Effect in Plasmonic Nanotip for LSPR Sensing (ID: sensors-690442)

Dear Editor:

Thank you so much for your editing and we are grateful for the reviewers’ comments concerning our manuscript (sensors-690442). These comments are all valuable and very helpful for the revision and improvement of our manuscript, as well as our research. We have considered the comments carefully and rewritten the contents to improve the quality of the manuscript and address the reviewers’ questions.

We are herewith submitting our point-by-point response to the reviewers’ comments. We hope these changes and answers can meet the requirements for the publication of our manuscript on Sensors. The detail of the revisions are as follows:

To Reviewer #1:

Comments: The question is interesting, however, the conclusions extend beyond what the data supports, i.e., "All of the above discussion implies that the very act of measuring the refractive index by use of LSPR actually changes the refractive index of the environment it is supposed to measure in the first place. This puts into question the whole enterprise of LSPR based instruments that are getting ubiquitous by the day for all sorts of measurements ranging from material identification to concentration measurement." These authors lump together the entire field of temperature-related LSPR studies with such general statements, however, the effects of photothermal generation are more far more nuanced depending on the light intensity and application needs.

Answer: The passage from the conclusion that you marked has been removed from the conclusion and has been changed as: “All of the above discussion implies that using a metal-cladded nanotip as an LSPR sensor requires extreme care as a mismatch of the physical dimensions of the nanotip, incident power and the length of the nanotip immersed in the solution can easily lead to increase in surrounding temperature causing inaccurate LSPR based sensing. On the hand, it also opens up the possibility of using a metal cladded nanotip as a low powered nanoheater. It would provide localized heating to only the desired location and the ability to control the location that is heated, something which is not possible with distributed nanoparticles used in most of the thermoplasmonic studies”.

Question 1: How generalizable are these findings to other nanostructures? Please comment 

Answer: The question of generalizability of the results has now been addressed in the manuscript’s discussion section by adding the following passage at line 270:

As the increase in temperature is caused by the interaction of surface plasmons with the metal [49–51]. This means that any situation which entails sufficient amount of such interactions should, in principle, lead to increase in temperature; causing temperature-based change in refractive index leading to inaccurate measurements of LSPR. For a given system, the increase in temperature (as clear from figure 4b and figure 4c given above) linearly depends on the input power of the laser source. This implies that in order to obtain accurate results from LSPR based instruments, the input laser power should be low enough to not to cause “significant” increase in the temperature. Given that our experimental design employs transmission spectroscopy, figure S3 shows that the reduction of power below a certain level makes the spectrum very noisy. In an experimental design that uses reflection spectroscopy, as in the case of [40], the resulting spectrum would be less noisy. This should easily enable them to further reduce the power. But this privilege is not available for an experimental design that uses transmission spectroscopy as used in this study.

In order to ascertain what can be considered as “significant” increase in temperature, the rate of change of refractive index with respect to temperature for the analyte also needs to be taken into account. It is because of the fact that the rate of change of refractive index with respect to temperature is not the same for all analytes, as given in figure 4a. This means that what is considered as a “significant” increase in temperature for a particular analyte, may or may not be “significant” for other analytes. Therefore, it is also an important parameter to be considered when designing a LSPR based sensing system

Question 2: Please add more discussion about how the light intensity relates to the values typically used in photothermal generation vs. sensing applications. The intensity and light source is often quite different between these two application cases and please provide more detailed information, preferably a table comparing key parameters so that it is clear to the readership

Answer: The discussion regarding intensity’s relation to sensing applications and thermoplasmonics has been added to the discussions section and the excitation intensity used in sensing studies and thermoplasmonic studies have been tabulated in table S3 and S4.

Question 3: In addition to photothermal generation, temperature-dependent LSPR sensing experiments have been conducted and should be discussed within this article, including aspects such as temperature on signal-to-noise ratio etc. Relevant references to discuss include: DOI: 10.1021/jp409264w and DOI: 10.1021/acs.analchem.7b03921.

Answer: The references you suggested have been incorporated in the manuscript.

Question 4: It is mentioned that "We have verified the presence of photothermal effect at the apex of a plasmonic tip. We explored its link with the change in the refractive index which is practically reflected in the shift of LSPR". This is too simplistic and bulk refractive index and surface sensitivities should be further analyzed in order to provide deeper and stronger quantitative insight.

Answer: The passage has been changed as: “We have looked at the presence of photothermal effect at the apex of a plasmonic tip. We explored its link with the change in the microscale refractive index in the vicinity of the nanotip. The temperature-based change in the refractive index is reflected by the shift of LSPR.”.

We hope the answers and revisions can meet the comments and thank you for your consideration.

Best regards,

Xiangwei Zhao

Reviewer 2 Report

This paper reports experimental and theoretical efforts for the study of thermal effects and refractive index change due to localized surface plasmon resonance (LSPR) in the vicinity of a gold nano tip. The authors observe plasmon resonance shift of the tip submerged in some common chemicals. There are several issues that need to be addressed and the manuscript requires dramatic improvements on the presentation in their data, figures, text and interpretations.

Important information regarding the method of FEM simulation like mesh size, boundary condition, excitation and polarization direction etc. should be mentioned in the manuscript. Proper validation of the simulation should be provided. Specifically, several spikes in the electric field intensity plot in figure S4 seems be due to problem in the simulation (possibly due to low mesh quality at the apex). A zoomed-in figure of the meshed geometry near the apex should be provided in the supporting. Also, in figure S4 the x-axis label – please use the symbol ‘µ’ and not the letter ‘u’. Towards the end of the paper the author talks about the effect of tip diameter on the electric field intensity and temperature, but the result of the analysis is not shown. For the experiments, the change in refractive index (RI) with temperature is very small and highly local (ranges within a few tens of micrometers). It is hard to believe that such minute change will give rise to a detectable shift in the LSPR peak. Also, what is the resolution of the grating used in the spectrometer. Proper control experiments are missing. The authors should repeat the experiment with a fiber tip of the same dimension but without the gold coating and show that the shift is due to LSPR only. Also, key information of the experimental setup is missing like working distance between the tip and the objective. Figure 5 does not have the propanol data. Also, the label is wrongly marked as figure 1 instead of figure 5. Authors mention that the rate of change of RI of water with respect to temperature is higher than the other organic molecules, but in figure 5 the peak shift in water is the least, which is contradictory since LSPR peak shift has a direct relationship with RI. A simulation graph corresponding to the experimental graph (LSPR peak shift vs intensity of excitation) in figure 5 should be provided. Title mentions ‘sensor’ but no performance evaluation of a sensor is made in the manuscript. What are the detection limit and the linearity range of the gold tip? These are very important parameters to evaluate the quality of a sensing technique. I recommend the authors to perform more experiments and provide a series of results on this. The figures in the paper requires a lot of improvement. For example, in figure 3a the scale bar is incomplete- mentioning only one point in the middle beats the purpose of a scale bar. The paper is poorly written with silly mistakes in several places. To name a few, figure 3 is wrongly mentioned as figure 1, unnecessary capitalization of words in between a sentence, inconsistency in terminology – ND-filter/ND filter, inconsistency in unit – sometimes space between number and unit sometimes not- please follow a consistent convention. Language requires lot of improvement. Introduction requires improvement. Many relevant literatures in the field is not cited, for example - Chen, Qin, et al. Nanomaterials12 (2017): 416, Jeon, Hui Bin, et al. Scientific reports9.1 (2019): 1-8, Jin, Xiulong, et al. Journal of Applied Physics 125.7 (2019): 073102,  etc.

Author Response

Prof. Dr. Xiangwei Zhao

State Key Laboratory of Bioelectronics

School of Biological Science and Medical Engineering, Southeast University

2 Si Pai Lou, Nanjing 210096, China

Tel/Fax: +86-25-83795632

15-Jan-2020

RE: Photothermal Effect in Plasmonic Nanotip for LSPR Sensing (ID: sensors-690442)

Dear Editor:

Thank you so much for your editing and we are grateful for the reviewers’ comments concerning our manuscript (sensors-690442). These comments are all valuable and very helpful for the revision and improvement of our manuscript, as well as our research. We have considered the comments carefully and rewritten the contents to improve the quality of the manuscript and address the reviewers’ questions.

We are herewith submitting our point-by-point response to the reviewers’ comments. We hope these changes and answers can meet the requirements for the publication of our manuscript on Sensors. The detail of the revisions are as follows:

To Reviewer #2:

Question 1: Important information regarding the method of FEM simulation like mesh size, boundary condition, excitation and polarization direction etc. should be mentioned in the manuscript. Proper validation of the simulation should be provided. Specifically, several spikes in the electric field intensity plot in figure S4 seems be due to problem in the simulation (possibly due to low mesh quality at the apex). A zoomed-in figure of the meshed geometry near the apex should be provided in the supporting.

Answer: The information regarding mesh has been added in the supplementary section (line 54) where simulation set up is explained, while the picture of meshed structure has been added in Figure S1 along with zoomed in figure of the apex. The information regarding boundary conditions and polarization direction have been tabulated in Table S2.

Question 2: In figure S4 the x-axis label – please use the symbol ‘µ’ and not the letter ‘u’.

Answer: The axis label for figure S4 has been corrected.

Question 3: Towards the end of the paper the author talks about the effect of tip diameter on the electric field intensity and temperature, but the result of the analysis is not shown.  

Answer: The details of output at the apex with respect to changes in fiber diameter and tip radius have been removed as the information was not useful in the explanation of the effect and a source of confusion.

Question 4: The change in refractive index (RI) with temperature is very small and highly local (ranges within a few tens of micrometers). It is hard to believe that such minute change will give rise to a detectable shift in the LSPR peak.

Answer: LSPR is actually very sensitive to very local refractive index of the medium. So, in case the local refractive index of the medium changes, it would lead to a shift in the LSPR peak.

Question 5: What is the resolution of the grating used in the spectrometer and distance between the tip and the objective?

Answer: The resolution of grating and the distance are 0.1 nm and approximately 2 cm respectively and we, have added them at line 127 and 128 correspondingly.

Question 6: Proper control experiments are missing. The authors should repeat the experiment with a fiber tip of the same dimension but without the gold coating and show that the shift is due to LSPR only.

Answer: The control experiment with IPA and its corresponding graph has been added as figure 5b in the manuscript.

Question 7: Figure 5 does not have the propanol data. Also, the label is wrongly marked as figure 1 instead of figure 5

Answer: IPA is short for Iso-Propyl Alcohol which is a type of propanol. The name has also been added in the manuscript in order to avoid the confusion.

Question 8: Authors mention that the rate of change of RI of water with respect to temperature is higher than the other organic molecules, but in figure 5 the peak shift in water is the least, which is contradictory since LSPR peak shift has a direct relationship with RI.

Answer: At line 186 of the manuscript, we say that “change of refractive index of water per unit temperature change is an order of magnitude smaller than the organic chemicals”. This explains the least shift in LSPR peak of water.

Question 9: A simulation graph corresponding to the experimental graph (LSPR peak shift vs intensity of excitation) in figure 5 should be provided

Answer: In the short term, we could not provide it. Our model of the fiber is large, that is why the simulations that we have shown in the manuscript are 2D-axisymeteric. In order to provide a graph of LSPR according to the response given in figure 5a, we need a 3D model which would require very large computational power, which we cannot afford in the short term.  

Question 10: Title mentions ‘sensor’ but no performance evaluation of a sensor is made in the manuscript. What are the detection limit and the linearity range of the gold tip? These are very important parameters to evaluate the quality of a sensing technique.

Answer: The title has been changed to “Photothermal Effect in Plasmonic Nanotip for LSPR Sensing”. The detection limit of LSPR sensors used in the literature has been added to the manuscript. But the detection limit of the nanotip we used is not mentioned because the focus of the manuscript is towards the fact that the nanotip is not a good sensing method for LSPR sensing.

Question 11: In figure 3a the scale bar is incomplete- mentioning only one point in the middle beats the purpose of a scale bar.

Answer: The figures have been improved as directed by the reviewer. And the problem with figure 3 specifically mentioned has been rectified.

Question 12: The paper is poorly written with silly mistakes in several places. To name a few, figure 3 is wrongly mentioned as figure 1, unnecessary capitalization of words in between a sentence, inconsistency in terminology – ND-filter/ND filter, inconsistency in unit – sometimes space between number and unit sometimes not- please follow a consistent convention. Language requires lot of improvement. Introduction requires improvement.

Answer: We have proofread the paper and tried to improve the language. We have tried to remove the unnecessary capitalizations and inconsistency of the terminology. The introduction section has been improved as well.

Question 13: Many relevant literatures in the field is not cited, for example - Chen, Qin, et al. Nanomaterials12 (2017): 416, Jeon, Hui Bin, et al. Scientific reports9.1 (2019): 1-8, Jin, Xiulong, et al. Journal of Applied Physics 125.7 (2019): 073102,  etc.

Answer: The literature you suggested has been added to the manuscript.

We hope the answers and revisions can meet the comments and thank you for your consideration.

Best regards,

Xiangwei Zhao

Reviewer 3 Report

Thermal effect is an intriguing problem in the applications of plasmonic sensors including SPR, LSPR, SERS and TERS, since plamson resonance is close related with heat generation due to the impedence of metals. Especially in the case of refractive index (RI) changes based sensors, the temperature effect should be taken care of. This paper addressed the problem from both simulation and experiment, which is very interesting and meaningful for researchers in the field of plasmonic sensor. The results and conculsion is solid to remind that in the plasmonic sensing of RI, the shift of resonance peak maybe derives from the tempetature but not the concentration. I recommend it to be accepted after minor reversion, and some suggestions are as follows:

The English of the writing should be improved and there are minor errors should be revised, for example, in Fig2b, the "eff" should be in lower case. Some latest good results related to LSPR-based sensors should also be mentioned by authors in the introduction besides some representative works, such as DOI: 10.1021/acs.jpclett.9b01390; DOI: doi.org/10.1364/OE.27.009879

Author Response

Prof. Dr. Xiangwei Zhao

State Key Laboratory of Bioelectronics

School of Biological Science and Medical Engineering, Southeast University

2 Si Pai Lou, Nanjing 210096, China

Tel/Fax: +86-25-83795632

15-Jan-2020

RE: Photothermal Effect in Plasmonic Nanotip for LSPR Sensing (ID: sensors-690442)

Dear Editor:

Thank you so much for your editing and we are grateful for the reviewers’ comments concerning our manuscript (sensors-690442). These comments are all valuable and very helpful for the revision and improvement of our manuscript, as well as our research. We have considered the comments carefully and rewritten the contents to improve the quality of the manuscript and address the reviewers’ questions.

We are herewith submitting our point-by-point response to the reviewers’ comments. We hope these changes and answers can meet the requirements for the publication of our manuscript on Sensors. The detail of the revisions are as follows:

To Reviewer #3:

Comments:

This paper addressed the problem from both simulation and experiment, which is very interesting and meaningful for researchers in the field of plasmonic sensor. The results and conculsion is solid to remind that in the plasmonic sensing of RI, the shift of resonance peak maybe derives from the tempetature but not the concentration. I recommend it to be accepted after minor reversion.

Question 1: The English of the writing should be improved and there are minor errors should be revised.

Answer: We have proofread the paper and tried to improve the language.

Question 2: . Some latest good results related to LSPR-based sensors should also be mentioned by authors in the introduction besides some representative works, such as DOI: 10.1021/acs.jpclett.9b01390; DOI: doi.org/10.1364/OE.27.009879

Answer: The literature you suggested has been added to the manuscript.

We hope the answers and revisions can meet the comments and thank you for your consideration.

Best regards,

Xiangwei Zhao

Round 2

Reviewer 1 Report

The authors have nicely revised the manuscript to incorporate key points and clearer discussion. I now recommend acceptance.

Reviewer 2 Report

The authors have significantly improved the manuscript and addressed most of the concerns. I recommend publication with further improvement of the English.